# From Monitoring to Prediction: Velocity-Based Strength Training in Female Floorball Athletes

**DOI:** 10.3390/sports13060175

**Published:** 2025-05-31

**Authors:** Basil B. Achermann, Naire Regazzi, Rahel Heynen, Dennis Lüdin, Julia Suter, Anna Drewek, Silvio R. Lorenzetti

**Affiliations:** 1Institute for Data Analysis and Process Design, ZHAW, 8400 Winterthur, Switzerlandlort@zhaw.ch (S.R.L.); 2Institution for Biomechanics, ETH Zurich, 8006 Zurich, Switzerland; 3Department of Health Science and Technology, ETH Zurich, 8006 Zurich, Switzerland; 4Section Performance Sport, Swiss Federal Institute of Sport Magglingen (SFISM), 2532 Magglingen, Switzerland

**Keywords:** velocity-based training, lower limb performance, female athletes, floorball, neuromuscular adaptations

## Abstract

This study examined the use of linear regression models for predicting the outcomes of a six-week velocity-based training (VBT) intervention in female floorball players. The intervention was integrated into regular training and consisted of brief 30-min sessions focused on back squats and trap bar deadlifts. Key performance metrics included neuromuscular adaptation, sprint speed, jump performance, stop-and-go (SAG) performance, and load-velocity profiles. Seventeen participants completed 12 training sessions, a strength block set (Sessions 1–6) and a power block set (Sessions 7–12). The predictive models explained 54% to 79% ($R^{2}$ = 0.54–0.79) of the performance improvement in the strength-related tests. Significant gains were observed in neuromuscular metrics, including estimated one-repetition maximum (1${\mathrm{RM}}_{est}$) and average mean concentric velocity for both exercises. These findings underscore the predictive potential of VBT in enhancing strength and power while highlighting the need to integrate task-specific exercises to optimize sport-specific performance. This study provides valuable insights for tailoring VBT strategies for female athletes in high-demand team sports such as floorball.

## 1. Introduction

In team sports, resistance training (RT) is critical for optimizing performance, preventing injuries, and developing sport-specific skills [1]. However, integrating RT into the demanding schedules of team sport athletes is challenging, as coaches must balance physical conditioning, recovery, and skill development to maximize benefits while avoiding overtraining.

Floorball, a sport known for its high intensity and dynamic demands, underscores the importance of efficient training methods. The sport requires rapid directional changes, sprints, and frequent stopping and starting, placing substantial demands on lower limb strength and neuromuscular control [2]. This need is illustrated by the high incidence of ankle and knee injuries among floorball players [3], with female athletes particularly vulnerable to such injuries due to their increased susceptibility to anterior cruciate ligament lesions [4]. Enhancing lower limb strength is crucial for improving performance and reducing injury risks among floorball players.

Monitoring RT is essential, as an athlete’s daily physical and psychological readiness can vary significantly, influencing how effectively preplanned training is executed regarding intensity and volume [5]. Regular monitoring provides critical information that enables coaches and athletes to better align training demands with the athlete’s current ability level, leading to more informed decisions [6]. Velocity-based training (VBT) has emerged as a promising and effective approach to resistance training. By using the speed of weight movements to guide the prescription of loads, sets, and repetitions, VBT can be used to dynamically adjust training loads based on an athlete’s performance [7].

Studies have shown that VBT can produce superior results compared to traditional weight training [8,9,10], often achieving these outcomes with reduced training volumes [8]. In addition, the immediate feedback provided during VBT sessions has been shown to boost athlete motivation and engagement, potentially contributing to greater performance gains [7,11]. Given its flexibility, VBT is particularly well-suited for resistance exercises that develop lower limb strength and power, such as squats and deadlifts [12]. While VBT has demonstrated benefits in improving maximal strength, jump performance, and sprint speed [13,14], much of this research has predominantly focused on males. Consequently, there is a knowledge gap regarding VBT’s effectiveness in females.

Notable features of VBT include the ability to estimate the one-repetition maximum (1RM) from submaximal effort, thereby reducing the time and risks associated with frequent maximal strength testing, particularly for less experienced athletes and large groups [15], and the use of velocity zones to target specific strength outcomes [16]. In addition, monitoring the velocity of repetitions during training provides a reasonable estimate of the metabolic stress induced by resistance training [17]. Due to fatigue, velocity naturally declines within a set of repetitions, thereby offering a real-time indicator of neuromuscular fatigue and training intensity [18]. Hence, VBT may serve as an objective monitoring tool, providing coaches with valuable data to support informed decision-making [7]. With the advancements in modern equipment, tracking every training repetition for an entire team has become feasible [19]. However, research on the effects of velocity and load trends throughout a training intervention remains limited, particularly in predicting outcomes and aiding in the decision-making process.

To address these gaps, this study investigated two key hypotheses through an analysis of a six-week VBT program followed by female floorball players.

**Hypothesis** **1.**
*Predictive power of velocity and load trends in VBT.*


Hypothesis 1 proposed that trends in velocity and load guided by the velocity framework could be used to predict intervention outcomes. Maintaining a target velocity often requires load adjustments, suggesting that both factors contribute to the prediction. This study systematically tracked all repetitions during 30-min VBT sessions featuring back squats and trap bar deadlifts to assess the predictive value of several VBT variables.

**Hypothesis** **2.**
*Effectiveness of minimalist VBT intervention for novice female floorball athletes.*


Hypothesis 2 examined whether a minimalist VBT intervention (2 × 30 min per week) could improve strength without prior team-based RT experience. The evaluation focused first on velocity parameters during resistance training (average velocity across multiple loads and estimated 1RM), and second on sport-specific performance measures, including sprint speed, jump height, and stop-and-go (SAG) ability. The analysis aimed to provide practical insights into the feasibility and effectiveness of VBT for female athletes in team sports.

## 2. Materials and Methods

This longitudinal experimental study included 12 training sessions over six weeks. The effect of the VBT intervention on lower limb performance was assessed at the baseline (T1: pretest) and after the intervention (T2: posttest).

### 2.1. Participants

A total of 23 female participants were initially enrolled in the study, of which 17 completed the intervention (n = 17). The reasons for withdrawal included personal circumstances, conflicts with the tight schedule, or discomfort in executing the trap bar deadlift. The participant characteristics at baseline were as follows: body weight = 64.5 ± 9.1 kg, height = 1.69 ± 0.06 m, and age = 22.0 ± 3.7 years. The participants were recruited from a top-level Swiss floorball team and were required to meet specific inclusion criteria, including active membership in the team, being over 18 years old, and having previous experience in strength training (at least one year). Exclusion criteria were any recent injuries that could affect participation in the training or testing procedures. The participants were advised to maintain their usual diet, use of nutritional supplements, and sleep habits during the study. Informed consent was obtained from all participants, and the study was conducted in accordance with the Declaration of Helsinki and approved by the local Ethics Committee of Bern, Switzerland (Project ID: 2021-00403).

### 2.2. Velocity Framework

The training intervention consisted of a six-week VBT program, with participants completing two sessions per week for a total of 12 planned sessions. To account for any missed sessions, two reserve sessions were included, although the primary analysis focused on the 12 planned sessions. The six-week duration, corresponding to 12 sessions, was selected based on previous research showing that similar VBT interventions lasting six to seven weeks can induce measurable neuromuscular adaptations [10,20]. This length of intervention represents a practical compromise: it is sufficient to elicit strength-related gains while ensuring participant compliance and minimizing dropout, especially in applied settings with elite athletes. The intervention was divided into two distinct blocks to target specific neuromuscular adaptations. The first block (Sessions 1–6) emphasized strength development, with training loads adjusted to maintain the mean concentric velocity (MCV) within a range of 0.55–0.7 m/s. The second block (Sessions 7–12) shifted the focus to power development, with a target MCV range of 0.8–1.0 m/s [21]. The training data from one representative athlete are illustrated in Figure 1. This progressive block design allowed the participants to transition from strength-focused training to power-oriented training. The training sessions included two primary exercises: the back squat and the trap bar deadlift. The strength session always occurred before the normal floorball training. The athletes performed their usual warm-up (dynamic stretching, core stability, sprints, and jumps) followed by a specific warm-up comprising two preparatory sets per exercise. The warm-up loads were set at 30 kg and 50 kg for the back squat, with an alternative of 40 kg for three athletes, and 35 kg and 55 kg for the trap bar deadlift. The load used for the second warm-up set also served as the minimum load for the main training sets. The participants performed the main training sets after the warm-up. During the strength block, these comprised three sets of five repetitions, while the power block consisted of three sets of eight repetitions. Velocity was the primary target for load adjustments: if three or more repetitions in a strength set, or five or more in a power set, fell outside the target velocity range, the load for the next set was adjusted by ±5 kg. Movement velocity was continuously monitored using the GymAware PowerTool, a tool tested for high validity and reliability in multiple studies [19,22], providing real-time feedback to ensure that the training loads aligned with the target velocity zones for each block.

### 2.3. Training Trend

For each training block (strength and power), robust regression (lmrob, R Version 4.4.1) was employed to analyze the trends in MCV (median) and training load (mean) for each participant. These trends, denoted as *MCV_STRENGTH*, *MCV_POWER*, *LOAD_STRENGTH*, and *LOAD_POWER*, (see Table 1) represented participant compliance with the prescribed velocity framework and captured training-related improvements or declines in performance. A compressed overview of all participating athletes is shown in Figure 2.

### 2.4. Testing Procedure

Assessments were conducted at T1 and T2 to evaluate key lower limb performance metrics. These assessments included 20 m sprint performance, stop-and-go (SAG) ability, jump performance through a countermovement jump (CMJ), and a squat jump (SJ). The test procedures have been described in the Swiss Olympic performance testing manual [23]. In addition, load–velocity profiles were assessed using the average MCV values across all repetitions executed and the estimated one-repetition maximum (1RM_est_); further details are provided in Appendix A and variables of interest in Table 1. The load-velocity profiles from T1 were also used to determine individualized starting loads to ensure an appropriate training intensity for each participant. Before testing, the participants completed a self-selected warm-up routine that included aerobic exercises, joint mobilization, and dynamic movements, similar to their regular floorball training preparation. Verbal encouragement was provided throughout to ensure maximal effort during the assessments.

### 2.5. Statistical Analysis

Statistical analysis was performed using the R environment [24]. Training progression was summarized by the slopes calculated from the robust regression (see Section 2.3) and was tested twice: (1) for differences and (2) for positive slope. The first test (1) was performed with respect to exercise type and power/strength (training type) using the Friedman rank sum test followed by post-hoc analysis. The second test (2) employed several one-sided Wilcoxon signed-rank tests. All *p*-values were corrected for multiple testing using the false discovery rate.

In the second step, the slopes representing the training progression were used to construct four linear regression models to predict improvements in velocity and 1RM (see Section B.1 for the model equations). Performance assessment was evaluated using linear mixed-effects models. This models accounted for repeated measurements per participant with a random intercept and contained body weight as a control variable. The model assumptions and potential interactions were checked using residual analysis. The significance level was set at p≤0.05. Effect sizes were classified according to Cohen [25] as small (≥0.2), medium (≥0.5), or large (≥0.8). Custom-written R script and associated dataset are available on the Open Science Framework repository (URL: https://osf.io/be85d/ accessed on 22 May 2025).

## 3. Results

### 3.1. Training Trend

There was a significant effect of conditioning on the slope of the MCV (Friedman rank sum test, $\chi^{2}\left(3\right)$ = 15.57, *p* = 0.001). Post hoc comparisons indicated that the slope for the Deadlift (*MCV_POWER*) differed significantly from the Deadlift (*MCV_STRENGTH*) (p=0.032) and the Squat (*MCV_STRENGTH*) (p=0.020). No other pairwise differences were significant (p>0.05).

Similarly, the slope of load varied significantly across conditions ($\chi^{2}\left(3\right)=23.46$, *p* < 0.001). Post hoc analyses showed that the Deadlift (*LOAD_POWER*) differed significantly from both the Squat (*LOAD_POWER*) (p=0.022) and the Squat (*LOAD_STRENGTH*) (p=0.001). There were additional significant differences between the Squat (*LOAD_POWER*) and Deadlift (*LOAD_STRENGTH*) (p=0.022), Squat (*LOAD_POWER*) and Squat (*LOAD_STRENGTH*) (p<0.001), and Deadlift (*LOAD_STRENGTH*) and Squat (*LOAD_STRENGTH*) (p=0.031). The only non-significant contrast was between Deadlift (*LOAD_POWER*) and Deadlift (*LOAD_STRENGTH*) (p=0.150).

For MCV, only the Deadlift (*MCV_POWER*) exhibited a slope significantly greater than zero (p=0.014), while there were no significant effects for the Squat (*MCV_POWER*) (p=0.180), Deadlift (*MCV_STRENGTH*) (p=0.813), or Squat (*MCV_STRENGTH*) (p=0.813).

For load, all conditions except for Squat (*LOAD_POWER*) showed a slope significantly greater than zero. Significant effects were observed for Deadlift (*LOAD_POWER*) (p=0.014), Deadlift (*LOAD_STRENGTH*) (p=0.014), and Squat (*LOAD_STRENGTH*) (p<0.001), while the Squat (*LOAD_POWER*) did not reach significance (p=0.727). The slopes are illustrated in Figure 3.

### 3.2. Predictive Models

Four linear models were constructed as predictors of the improvements in velocity and estimated 1RM. Table 2 provides an overview of these four models, while Figure 4 illustrates their predictive performance. The complete model specifications are detailed in the Section B.1, with comprehensive results presented in Section B.2. Several significant predictors for velocity differences (*AVDIFF*) and estimated 1RM differences (*RMDIFF*) between Test 1 (T1) and Test 2 (T2) were identified.

#### 3.2.1. AVDIFF Models

**Squat**: The predictor *AVT1* was significantly negatively associated with *AVDIFF* (β=−0.793,SE=0.223,p<0.01). Additionally, the predictor *MCV_POWER* showed a trend within the 10% significance threshold, suggesting potential relevance. The model fit was improved by 14% compared to the baseline model.**Deadlift**: *AVT1* was significantly and negatively associated with *AVDIFF* (β=−0.541, *SE* = 0.221, *p* < 0.05). Furthermore, *LOAD_STRENGTH* was significantly positively related (β=0.036,SE=0.012,p<0.05), indicating that higher loads were associated with greater differences in velocity. The model fit was improved by 22% compared to the baseline model.

#### 3.2.2. RMDIFF Models

**Squat**: *MCV_POWER* (β=−870.416,SE=332.681,p<0.05) and *MCV_STRENGTH* (β=855.997,SE=305.569,p<0.05) were significant predictors, highlighting the importance of the slope of velocity in estimating differences in 1RM. *RMT1* demonstrated a strong negative relationship with *RMDIFF* (β=−0.89,SE=0.177,p<0.001), emphasizing its critical role in the estimation of 1RM. The model fit was improved by 17% compared to the baseline model.**Deadlift**: Only *BMT1* was significant (β=1.030,SE=0.372,p<0.05), suggesting that body mass influenced differences in 1RM. *MCV_STRENGTH* showed a potential effect within the 10% significance threshold. The model fit showed marginal improvement (2%) compared to the baseline model.

Given the relatively small sample size, the observed trends approaching significance warrant cautious interpretation and further investigation.

### 3.3. Testing

The results of the T1 and T2 assessments are summarized in Table 3. *Sprint and SAG:* No significant changes were observed in sprint performance. The 5 m sprint time increased slightly by 1.18% (p=0.593), while the 20 m sprint showed a marginal, non-significant improvement (p=0.130). In contrast, the SAG test showed a small but statistically significant improvement (Δ=−0.04 s, p=0.025, Effect size = −0.36, small). *Jumps:* In the CMJ, relative power increased significantly by 2.1% (p=0.048), whereas jump height did not change (p=0.464), with both showing small effect sizes. In the SJ, both relative power (4.8%, p<0.001) and jump height (3.9%, p=0.027) improved significantly, with a large effect size for relative power and a small effect size for height. *Load–Velocity Profiles:* Significant improvements were observed in both the squat and trap bar deadlift. Average velocity (AVDIFF) increased in the squat (p=0.016, Effect size = 0.61, medium) and trap bar deadlift (p=0.002, Effect size = 0.82, large). Estimated 1RM relative to body weight (RMDIFF) also increased significantly: by 15.2% in the squat (p=0.001, Effect size = 0.74, medium) and by 5.9% in the trap bar deadlift (p=0.006, Effect size = 0.60, medium). Additionally, body weight increased significantly over the intervention period (mean change = 2.15%, Δ=1.39 kg, p<0.001).

## 4. Discussion

In this study, we tested two hypotheses: (1) that velocity and load trends within a velocity framework could predict training outcomes, and (2) that a minimalist VBT program could improve strength while being effectively integrated into team training. In the following discussion, we first address the observed trends in training shaped by the velocity framework. We then evaluate the predictions of the regression models. The feasibility and effectiveness of the intervention are considered within the constraints of elite team sports. Lastly, we discuss the applied velocity framework, including potential adjustments to further optimize its effectiveness.

### 4.1. Training Trend

The main difficulty in the analysis of training data and the interpretation of trends lies in its association with numerous factors, including training adaptation, compliance, neuromuscular fatigue, technical proficiency, and external feedback, among others [7]. For the results to serve effectively as an objective monitoring tool, they must reliably represent training adaptation and neuromuscular fatigue. When the results meet these criteria, they can inform training decisions, potentially offering professional athletes a competitive advantage. Figure 3 presents a visualization of the results.

Our study revealed pronounced differences in load trends between the strength and power training blocks. The strength block exhibited greater variability, with most participants showing a strong positive trend, indicating that velocity was primarily modulated through progressive load increments across sessions. In contrast, the power block demonstrated lower inter-participant variability, with load trends clustering just above zero. This was likely due to the velocity framework constraining load adjustments and requiring the participants to maintain a consistent load. The higher repetition count (eight repetitions in the power block), which had to be completed within the designated velocity zone, was most likely the primary factor limiting load modulation. In addition, the minimum prescribed load was set at the last warm-up set, forcing some participants to act in order to reach the velocity zone rather than simply remaining within it. This approach was intentional, ensuring a minimum load stimulus while encouraging participants to meet the targeted velocity ranges.

While the load progression differed between training blocks, velocity trends followed a distinct pattern. Except for the deadlift in the power block, the slopes for the velocity trends were not significantly greater than zero. In the strength block, the velocity trends were generally negative, whereas in the power block, they were predominantly positive. These results align with the observed load trends, supporting the observation that during the strength block, the participants prioritized progressive increases in the load while maintaining their velocities within the prescribed zones. This interplay between load and velocity progression underscores its potential for predictive modeling while also emphasizing the influence of the velocity framework and its prescribed zones. The observed trends were likely specific to this framework and could differ under alternative scenarios.

### 4.2. Model Performance

Statistical models should be interpreted with caution when generalizing findings. Models represent the relationship between factors measured in a controlled and specific environment and must be altered accordingly when applied to different conditions [26]. However, once validated and supported by empirical evidence, such models might serve as valuable tools in the decision-making process, aiding athletes and coaches to optimize training strategies. The models presented in this study were based on a sample of 17 participants and thus are specific to the applied velocity framework that included 12 training sessions, predefined velocity zones for the strength and power blocks, and a structured set of rules guiding load adjustments. Adherence to these velocity zones likely had the greatest impact on load and velocity trends, along with individual adaptations to training. Four regression models (see Section B.1) were developed using the training data to predict the results in the form of AVDIFF and RMDIFF for the squat and deadlift. Model 1 (AVDIFF Squat) and Model 2 (RMDIFF Squat) exhibited the highest predictive power (Adjusted $R^{2}$ = 0.786 and 0.701), while the deadlift models (Models 3 and 4) had lower explanatory power (Adjusted $R^{2}$ = 0.617 and 0.544; see Figure 4). Models 1 to 3 demonstrated improvements in the fit of 14% to 23% compared to the baseline models, suggesting their potential utility for further analysis. In contrast, Model 4 showed no significant improvement. Due to the small sample size, model parameters with a significance level of p<0.1 are included in the discussion but should be interpreted with caution.

In **Model 1 (Squat, AVDIFF)**, a higher initial absolute strength (AVT1) was a significant predictor of a smaller improvement in velocity (p<0.01), indicating that faster and stronger individuals demonstrated less change in movement velocity. In addition, MCV_POWER showed a trend toward significance (p<0.1), suggesting a predictive value of the velocity trend in the power block to velocity adaptations.

In **Model 2 (Squat, RMDIFF)**, RMT1 was a strong negative predictor of 1RM improvement (p<0.001), consistent with the observation that participants with lower initial strength levels experienced greater relative gains. Both MCV_STRENGTH and MCV_POWER were significant for improvement in 1RM (p<0.05), highlighting the relevance of velocity trends across different load zones. LOAD_POWER had a weaker effect (p<0.1).

In **Model 3 (Deadlift, AVDIFF)**, AVT1 again negatively influenced improvement in velocity (p<0.05), supporting the squat findings. LOAD_STRENGTH (p<0.05) and body mass (BMT1; p<0.05) were significant predictors, emphasizing the roles of load exposure and body mass in the results for deadlift velocity.

In **Model 4 (Deadlift, RMDIFF)**, BMT1 significantly predicted 1RM gains (p<0.05), whereas RMT1 showed no significant effect. The absence of a clear relationship between RMT1 and deadlift strength gains may reflect variability in measurement accuracy or technical execution. No clear trends were observed for velocity or load parameters in this model. Given the minimal improvement from baseline, the velocity and load trends did not provide additional predictive value in this model.

These findings lay the foundation for the development of more robust predictive models to support training decisions, using larger datasets. While based on a small sample, they highlight patterns that may assist in individualizing load progression. Higher initial strength consistently limited adaptation, suggesting that stronger individuals may require different loading strategies to induce further gains (Figure 4). Body mass influenced deadlift outcomes more than squat performance, reflecting its relevance in exercise selection and technique. Although velocity and load trends were inconsistent, they improved model fit in most cases, indicating potential value for monitoring training responses. With larger datasets, such models could be refined to guide individualized programming more effectively.

### 4.3. Neuromuscular Adaptations and Performance Outcomes

The VBT program led to clear improvements in neuromuscular responses, as demonstrated by the enhanced velocity and increases in estimated 1RM. These changes were consistent with prior research highlighting the role of VBT in maximizing neuromuscular adaptations [21,27,28,29,30]. However, task-specific performance outcomes, such as sprint ability, did not show significant improvement which was unexpected [14], and while SAG and jump performance improved significantly, the effect size was small to medium. This suggests that the transfer of strength gains to task-specific performance is limited. This may be attributed to the duration of the intervention, as 12 sessions may have been insufficient, or to a potential delayed transfer effect occurring two to four weeks post-intervention. Additionally, the absence of a control group limits the ability to isolate the effect of the VBT intervention itself. Including a comparison group undergoing conventional training would have strengthened causal inference and should be considered in future studies. In the SJ, relative power was improved with a comparatively large effect size; the weight showed a medium effect size, consistent with previous studies [10,27,28,29,30]. However, the effect sizes for the CMJ variables were lower than those reported in previous studies, and thus the lack of similar improvements was somewhat unexpected. This discrepancy highlights the complexity of gains in jump performance and suggests that further investigation into training parameters, such as load selection and exercise specificity, may be necessary to optimize training.

Interestingly, SJ metrics, including relative power and jump height, showed greater improvement compared to CMJ, with only a small absolute difference between the two tests. This contrasts with the typical pattern, where CMJ performance generally exceeds SJ performance [31]. A recent meta-analysis of nine VBT interventions in male athletes reported a moderate positive effect on 1RM (standardized mean difference = 0.76), a moderate effect on CMJ (standardized mean difference = 0.53), and a small but favorable effect on sprint performance (standardized mean difference = −0.40) [14]. Our findings in female athletes reflect a similar trend for strength-related outcomes, but showed less pronounced improvements in CMJ and no effect on sprint performance, which warrants further investigation.

The unusual CMJ vs. SJ outcome pattern may have been influenced by factors unrelated to training adaptation. The CMJ is more sensitive to neuromuscular fatigue than the SJ [32], and several participants competed in matches two days prior to post-testing, conditions that reflect real-world constraints when working with elite athletes. Additionally, the CMJ involves a stretch-shortening cycle, unlike the SJ and the primary training exercises, which may have introduced technical variation that biased results in favor of the SJ.

The limited transfer of strength gains to improvement in CMJ highlights the need for exercises that more accurately replicate the biomechanical and coordinative demands of elastic jumps. Research suggests that plyometric training, which enhances elastic energy utilization and neuromuscular coordination, is particularly effective for improving jump performance [33]. This suggests that a combination of training methods, such as integrating plyometrics with VBT, may be necessary to provide the specific stimuli needed to optimize jump performance. Additionally, Sprint and SAG performance showed no significant changes, likely due to the limited specificity of the VBT intervention [10]. These tasks rely on the production of horizontal force, rapid acceleration, and short ground contact times, factors that were not sufficiently stimulated. Rodríguez-Rosell et al. (2017) [34] demonstrated that combining light-load maximal velocity training with plyometrics enhances sprint performance more effectively than weight training alone, drawing the same conclusion as with the CMJ. Thus, a mixed training approach may be necessary for optimal transfer. While our study focused specifically on a minimalist VBT protocol, we acknowledge that a comparison group performing conventional strength training without VBT would have strengthened the interpretation of the observed effects. Future research should consider this design to better isolate the contribution of VBT-specific adaptations.

The three players with the lowest initial strength levels (Figure 2, Low Performance) demonstrated substantial improvements during the study. This aligns with the previously discussed trends and models, supporting the application of the implemented velocity framework in a team setting. In contrast, the participant with the highest initial strength level (Figure 2, High Performance) showed minimal improvement. Notably, distinguishing between low and high performers based solely on mean concentric velocity, particularly in the strength block, proved challenging.

A major advantage of VBT is the objective measurement of speed and power, providing immediate feedback that allows athletes to track their progress, compare performances, and adjust training accordingly. Despite the group-based nature of team sports, VBT enables individualized training within a team setting. Instant feedback can enhance motivation, encouraging athletes to push themselves as they observe their improvement. Moreover, coaching staff reported increased athlete confidence in resistance training throughout the intervention, with the athletes showing greater willingness to lift heavier loads without hesitation [35].

### 4.4. Limitations

Several limitations must be considered when interpreting the present findings. The small sample size (n = 17) and the absence of a control group limit the generalization of the results and preclude definitive causal inferences concerning the effectiveness of the interventions. In elite sports, it is difficult to implement a strength training program during a competition season and to find a large number of participants. Moreover, the study was not blinded, and both training and testing were supervised by the same staff. This may have introduced expectancy effects or unintentionally influenced effort levels. Participants performed self-selected warm-ups before testing, which may have introduced variability in neuromuscular readiness and affected performance outcomes. The power-focused training block overlapped with the floorball competition season. The cumulative fatigue from regular matches likely reduced recovery capacity and may have interfered with performance adaptations. However, no systematic effect of match timing was observed. Furthermore, no long-term follow-up was conducted. Without post-intervention monitoring, it remains unclear whether the observed strength gains were maintained during the competitive season, limiting conclusions about the sustained relevance of the intervention. However, the time-efficient nature of the program could allow for continued in-season implementation, which may be desirable in applied settings.

The design of the velocity framework offered valuable insights into the effectiveness of training parameters. The strength block, conducted during the preseason, led to greater load increments with constant velocity compared to the power block (Figure 3). The power block coincided with the competitive season, a factor that may account for the comparatively smaller improvements observed. The demanding schedule during the competitive season likely reduced the athletes’ capacity for recovery, limiting the adaptation to power-focused training. Despite these challenges, the intervention demonstrated feasibility and time efficiency, with athletes achieving notable improvements in strength through just two 30-min sessions per week, highlighting the practicality of the program within a demanding training schedule. The absence of a mid-point test and randomization limits the interpretation of adaptations within each training block. Future studies should consider intermediate testing to assess the time course and specific effects of the different training blocks. In addition, inconsistencies in exercise technique, particularly during squats, may have affected the transfer of gains in strength and power to improvement in performance. Variability in the range of motion or execution form could have led to uneven neuromuscular stimulus among participants despite efforts to standardize instructions and supervision. Future studies may consider implementing box squats to standardize the range of motion. These inconsistencies highlight the challenges of ensuring uniform technique across a group of athletes and the potential impact on the outcomes of training interventions. Furthermore, no actual 1RM testing was performed due to the athletes training history. As they had no prior experience with 1RM testing and lacked sufficient technical proficiency, a true 1RM assessment was considered unsuitable. This likely led to overestimation in the derived 1RM values (1RM_est_), as noted in the literature [20].

### 4.5. Practical Recommendations

Based on our findings, several practical recommendations can be made to enhance the effectiveness of VBT protocols for female floorball athletes. First, this study demonstrates the feasibility of integrating a VBT training intervention consisting of two 30-min sessions per week into a demanding training schedule. This highlights the potential for time-efficient resistance training programs to produce significant neuromuscular adaptations, even when conducted alongside regular team practices. Incorporating complementary plyometric exercises such as jumps, sprints, and directional changes could address the biomechanical and coordinative demands of tasks like sprinting and jumping, potentially further improving the transfer of strength gains to sport-specific performance.

Although limited by a small sample size, the modeling results provide preliminary evidence that initial strength level and velocity may influence training responses. For example, load progression in the strength block was associated with greater velocity gains in the deadlift, while consistent velocity trends were linked to improvements in squat 1RM. Some effects only approached statistical significance (p<0.1) and should be interpreted with caution. These trends, however, suggest potential directions for future work, particularly in identifying athletes who may benefit from specific loading strategies. Body mass also appeared to contribute to deadlift improvements, underscoring the multifactorial nature of RT adaptation. While the current findings are not definitive, they highlight the promise of velocity tracking for individualized monitoring, especially if replicated in larger samples. A clearly defined velocity framework may offer a foundation for developing more robust predictive tools to guide programming decisions in team sport environments.

Future research should investigate the impact of various training factors, including session duration, velocity and repetition ranges, and feature extraction methods. In particular, the six-week training period used in this study warrants further analysis, as a shorter window may be more effective for detecting early adaptations due to training and facilitating timely decision-making.

#### Potential Adjustments to the Velocity Framework

The following adjustments to the velocity framework may provide further insights and optimization:Definition and refinement of velocity zonesDuration between testing and prediction time framesNumber of repetitions per setThreshold of repetitions outside the target velocity zone before load adjustmentsMinimum load requirementsExercise selection

Further exploration of these factors may strengthen the accuracy and practical relevance of trend-based models in resistance training.

## 5. Conclusions

This work highlights the potential of using predictive models based on clearly defined velocity frameworks to anticipate training outcomes in VBT. Although developed on a small sample, the models demonstrated predictive ability and suggest that objectively monitoring training progress may help identify individual response patterns. Such approaches could help coaches and managers detect non-responders or athletes exhibiting negative trends, enabling timely adjustments to training programs. Future research should explore various training configurations to enhance the predictive accuracy of the models and extend their generalizability to larger populations. Investigating different velocity zones, repetition schemes, and training frequencies could further optimize training effectiveness and increase model precision.

The performance outcomes confirmed the practicality of incorporating minimalist yet effective resistance training sessions, even within the constraints of a competitive season. Two weekly 30-min sessions yielded significant improvement in neuromuscular responses, as reflected in enhanced load-velocity profiles, estimated one-repetition maxima, and vertical jump power. These results demonstrate the efficiency and applicability of this approach under the hard constraints of elite team sport schedules. However, despite notable improvements in neuromuscular performance, these enhancements did not consistently translate into significant gains in sport-specific tasks such as sprint speed or stop-and-go ability. This limited transfer highlights the necessity of integrating additional sport-specific and plyometric exercises that closely replicate the biomechanical and coordinative demands of competitive play. Future training programs should aim to balance general neuromuscular gains with targeted interventions that enhance task-specific athletic performance.

## Figures and Tables

**Figure 1 sports-13-00175-f001:**
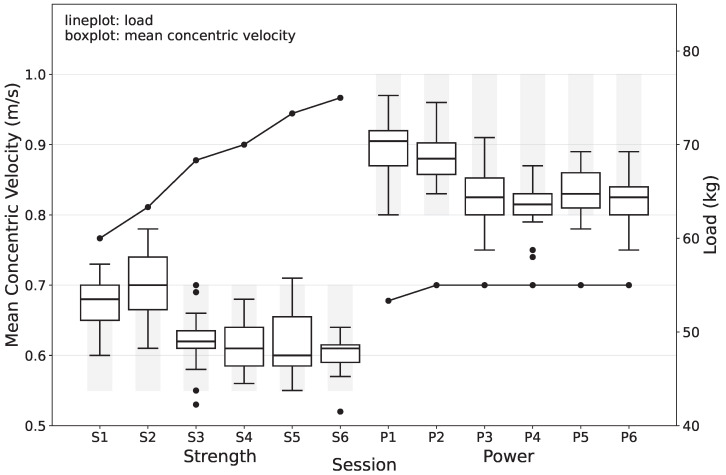
Representation of the velocity framework for one athlete. The boxplots illustrate the distribution of the mean concentric velocity across all repetitions (15 repetitions per session in the Strength block, 24 repetitions per session in the Power block). The black dots represent outliers. The shaded areas behind the box plots represents the velocity zones of the applied training framework. The line plots represent the average of the three selected loads per training session. A clear upward trend in load is evident in the Strength block, whereas the load is kept constant in the Power block to ensure adherence to the velocity zone.

**Figure 2 sports-13-00175-f002:**
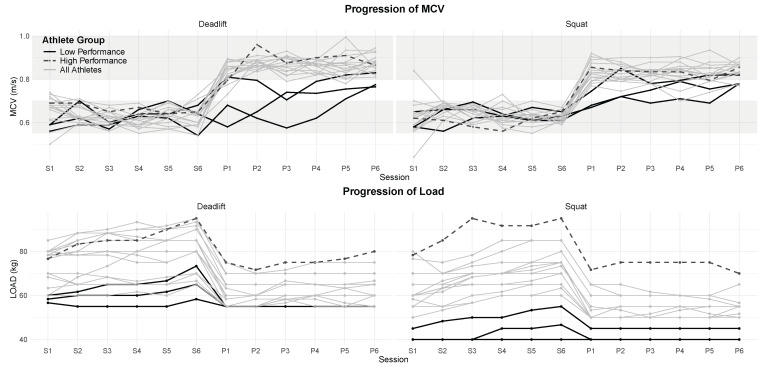
Training progression in Strength (S) and Power (P) blocks for the back squat and trap bar deadlift. The first row shows the progression in the mean concentric velocity (MCV); the second row shows load progression. The results for the Deadlift (**left**) and squat (**right**) are shown. The grey shading represents the prescribed velocity zones. The Low Performance group included the three athletes with the lowest initial strength levels, while the High Performance athlete represents the maximum of the strength spectrum.

**Figure 3 sports-13-00175-f003:**
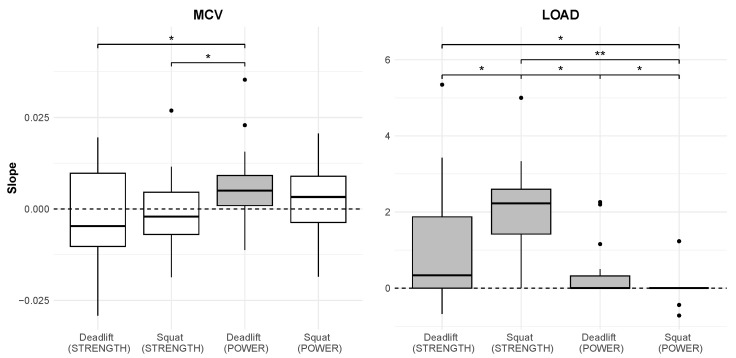
Boxplots illustrating the distribution of slopes derived from robust regression analyses across six training sessions for each block (Strength, Power) of the velocity framework. Significant differences are denoted as follows: * *p* < 0.05, ** *p* < 0.001. The grey backgrounds indicate slopes that are significantly greater than zero. MCV: Mean Concentric Velocity.

**Figure 4 sports-13-00175-f004:**
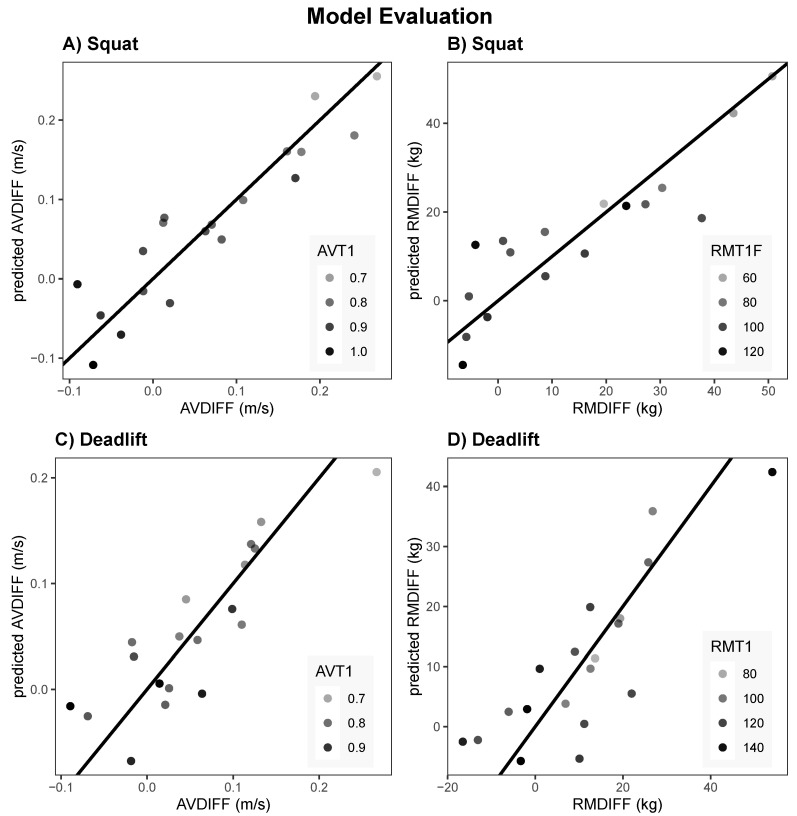
Visualization of model performance in predicting the average differences in velocity (AVDIFF: (**A**,**C**)) and estimated 1RM (RMDIFF: (**B**,**D**)). The first row represents model performance for the back squat data, while the second row corresponds to the deadlift data. The color coding indicates the initial average velocity (AVT1: (**A**,**C**)) and initial estimated 1RM (RMT1F: (**B**), RMT1: (**D**)).

**Table 1 sports-13-00175-t001:** Overview of key variables used for test evaluation, trend analysis and in the predictive models. T1: pretest, T2: posttest.

Variable	Unit	Description
**AVDIFF**	m $s^{-1}$	Velocity Difference of load–velocity profile
**RMDIFF**	kg	Estimated 1RM Difference
**AVT1**	m $s^{-1}$	Average MCV of load–velocity profile at T1
**AVT2**	m $s^{-1}$	Average MCV of load–velocity profile at T2
**RMT1**	kg	Estimated 1RM at T1
**RMT2**	kg	Estimated 1RM at T2
**BMT1**	kg	Body Mass at T1
**BMT2**	kg	Body Mass at T2
**MCV_POWER**	m $s^{-1}$ /session	Training trend of MCV (Power Block)
**MCV_STRENGTH**	m $s^{-1}$ /session	Training trend of MCV (Strength Block)
**LOAD_POWER**	kg/session	Training trend of Load (Power Block)
**LOAD_STRENGTH**	kg/session	Training trend of Load (Strength Block)

**Table 2 sports-13-00175-t002:** Summary of multiple linear regression models for the differences in velocity and estimated 1RM between T1 and T2 *.

Model	Variable	Exercise	RMSE	DF Residual	R-Squared	Baseline R-Squared	F-Statistic	*p*-Value
Model 1	AVDIFF	Squat	0.07	12	0.79	0.65	12.01	<0.001
Model 2	RMDIFF	Squat	16.8	10	0.67	0.5	6.34	0.006
Model 3	AVDIFF	Deadlift	0.08	12	0.62	0.39	5.83	0.005
Model 4	RMDIFF	Deadlift	14.0	12	0.54	0.52	4.58	0.012

* AVDIFF (Velocity) and RMDiff (Estimated 1RM) represent the differences between Test1 and Test2. DF Residuals: Degrees of Freedom. R-squared: adjusted R-squared values of the linear models. Baseline R-squared: adjusted R-squared values of the linear model based on baseline performance values (AVT1, RMT1) and BMT1. RMSE: Root mean squared error based on LOOCV.

**Table 3 sports-13-00175-t003:** Results pre and post testing *.

	T1	T2	Δ	Δ%	*p* Value	ES
**Sprint**						
5 m Time (s)	1.15 ± 0.06	1.16 ± 0.07	0.01	1.18	0.593	0.23
20 m Time (s)	3.41 ± 0.13	3.40 ± 0.13	−0.01	−0.33	0.13	−0.19
**SAG**						
Time (s)	5.01 ± 0.17	4.97 ± 0.19	−0.04	−0.7	**0.025**	−0.36
**CMJ**						
Max Height (cm)	30.2 ± 4.1	30.3 ± 3.3	0.2	0.6	0.464	0.09
Max Relative Power (W/BW)	46.8 ± 5.1	47.7 ± 5.8	1.0	2.1	**0.048**	0.37
**SJ**						
Max Height (cm)	28.3 ± 4.2	29.4 ± 3.5	1.1	3.9	**0.027**	0.47
Max Relative Power (W/BW)	44.4 ± 5.6	46.5 ± 5.4	2.1	4.8	**<0.001**	0.89
**Squat**						
AV (m/s)	0.88 ± 0.10	0.95 ± 0.06	0.07	7.5	**0.016**	0.61
1${\mathrm{RM}}_{est}$ (kg/BW)	1.44 ± 0.27	1.66 ± 0.34	0.22	15.2	**0.001**	0.74
**Trap Bar Deadlift**						
AV (m/s)	0.86 ± 0.09	0.92 ± 0.07	0.06	7.3	**0.002**	0.82
1${\mathrm{RM}}_{est}$ (kg/BW)	1.76 ± 0.30	1.87 ± 0.23	0.10	5.9	**0.006**	0.60

* Data are mean ± SD; T1: Pre-intervention testing; T2: Post-intervention testing; Δ: Absolute change; Δ%: Percentage change; p value: Statistical significance; ES: Effect size (Cohen’s d); SAG: Stop-and-Go; CMJ: Countermovement jump; SJ: Squat jump; AV: Average MCV against all absolute loads; V0: Theoretical maximal MCV at zero load; 1${\mathrm{RM}}_{est}$: Estimated one-repetition maximum relative to BW. The results are corrected for body weight.

## Data Availability

The dataset and analysis code scripts are available at the Open Science Framework (URL: https://osf.io/be85d/ (access on 22 May 2025)).

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
