# Peer review of "From Monitoring to Prediction: Velocity-Based Strength Training in Female Floorball Athletes"

_sports, 2025, doi:10.3390/sports13060175_

Round 1

Reviewer 1 Report

Comments and Suggestions for Authors

General Comments

  1. I found the topic both relevant and timely, especially given the growing attention to load monitoring and individualized strength training approaches in female team sports. Velocity-based training (VBT) and predictive modeling in a real-world setting with elite female floorball athletes is a commendable and much-needed contribution.
  2. That said, I have significant reservations regarding the overall strength of the study design. Most critically, the lack of a control group and the small sample size (n = 17) limit the reliability and generalizability of the findings. The authors make strong claims about predictive accuracy and training feasibility, but the data presented do not support these conclusions as robustly as suggested.
  3. The study is well-organized and supported by recent literature, but it often overstates its predictive models’ precision and practical implications. While the adjusted R² values are indeed promising, the sample is too small to validate these models in a statistically meaningful way.
  4. The intervention is well-structured and feasible for implementation in team sports contexts. However, real-world effectiveness remains limited and underexplored, particularly in transferring neuromuscular gains to task-specific outcomes such as sprinting and change of direction.
  5. I appreciated the statistical sophistication in constructing training trend slopes and linear models. Still, I question whether the models are being interpreted with sufficient restraint, especially given the study's exploratory nature. Claims about individual optimization and athlete classification (e.g., non-responders) should be treated more cautiously.

Specific Comments

  1. The introduction is comprehensive and well-referenced. The rationale for focusing on VBT in female athletes is justified. I particularly appreciated the effort to position the study within broader sport-specific and injury-prevention contexts. However, Hypothesis 2 (Lines 68–73) suggests effectiveness without defining success criteria beyond strength improvement. I would have expected a more cautious framing here.
  2. The training intervention is well described, including clear velocity targets and real-time load adjustments (Lines 92–116). However, the authors do not address the potential measurement bias introduced by the exclusive reliance on GymAware tools without cross-validation.
  3. The testing procedures and load-velocity profiling are methodologically sound (Appendices A.2–A.3), though using estimated 1RM (1RMest) rather than actual maximal testing is a limitation—especially in a performance-focused study. The justification for not including real 1RM testing should have been better articulated (Line 390).
  4. The approach to modeling slopes using robust regression, linear mixed-effects models, and FDR correction is technically appropriate. However, despite making predictions about complex adaptive phenomena, no justification is given for using linear models over non-linear or machine learning methods. Additionally, no cross-validation is performed, raising questions about the models' generalizability (Appendix B.2).
  5. The improvements in neuromuscular measures (e.g., 1RMest and velocity) are statistically significant and practically meaningful. However, I noticed that effect sizes for sprint and SAG tests were trivial or minor, which the authors do acknowledge (Lines 208–218), though somewhat too briefly.
  6. The model fit statistics (Adjusted R² = 0.54 to 0.79) are strong but should be interpreted cautiously given the degrees of freedom and absence of validation. The figures in Figure 4 visually reinforce the model claims, but the scatter shows considerable unexplained variance (Page 7).
  7. The discussion on training trend dynamics and model interpretation (Lines 229–310) is thorough and insightful but sometimes speculative. For instance, the statement that coaches can “detect non-responders” based on these predictive models (Line 437) seems overstated, given the limited sample and lack of testing beyond six weeks.
  8. I agree that neuromuscular gains did not transfer well to task-specific skills (Lines 311–344). The reasoning is plausible—insufficient specificity, volume, or inclusion of plyometrics—but these factors could have been controlled for. Including a comparison group doing similar strength training without VBT would have helped isolate the intervention effect.
  9. I was glad to see the authors discuss the unusual CMJ vs. SJ outcome pattern (Lines 328–334). However, they stopped short of exploring how fatigue, technical variation, or data noise may have confounded these measurements, which should have been part of the limitations section.
  10. The authors acknowledge the small sample size and lack of a control group but frame these issues as logistical rather than methodological. They should go further. For instance:
    • No blinding, which may have biased effort levels during training and testing.
    • Self-selected warm-ups before testing, which could have influenced neuromuscular readiness (Appendix A.2);
    • Concurrent floorball matches during the power block (Line 375) may have led to cumulative fatigue.
  11. The lack of long-term follow-up is also worth noting. If strength gains fade quickly or do not sustain into the competitive season, the intervention's practical relevance is questionable.

Reviewer 2 Report

Comments and Suggestions for Authors

The authors report efficacy and efficiency of a velocity-based training (VBT) protocol in female floorball players. The authors note that improved performance was modest and limited to a few measures like stop and go and jump performance, but not task-specific measures like sprint ability. The authors also develop regression-based models in attempts to predict improvements in velocity and 1RM. Unsurprisingly, greater initial strength and velocity were associated with smaller gains in those measures, which the authors attribute to the principle of diminishing returns.

Generally, the data are solid, but there are several weaknesses that should be addressed.

After citing previous studies regarding VBT, the author cites the lack of females in VBT studies as a knowledge gap. In their lengthy discussion the authors do not include any discussion of their findings in the context of previous findings in male subjects.

The authors should provide a rationale for implementing strength training before power training in all subjects instead of randomizing these blocks. How do the authors know that strength training did not influence power training, and how might this affect interpretation when comparing slopes between the two modalities? 

There are discrepancies between the results reported in section 3.1 and Figure 3. Lines 155-157 state that Deadlift (MCV_Strength) differed...from Squat (MCV_Strength). There are no indicators of this comparison in Figure 3. Instead, Figure 3 indicates a significant difference between squat strength and squat power. This difference is not included in Lines 155-157.

In lines 375-377 the authors speculate that power training could have been affected by its implementation during the competitive season. Is there variability in playing time between the study participants? If so, individual playing time data might shed some light on this. 

There are too many abbreviations in the manuscript. Some (LVP, FDR, LME, ES) are not used frequently enough to abbreviate, and some are defined more than once (LVP defined on lines 5, 71-72, and 129). Defining mean concentric velocity (MCV) and average mean concentric velocity (AV) is confusing. 

Overall, while the results are interesting, it is unclear how they will inform training decisions.

Reviewer 3 Report

Comments and Suggestions for Authors

If you use BW as the relative load reference, you should apply that terminology consistently throughout the manuscript, including in the SJ and CMJ table.

Why was the intervention limited to only 6 weeks?

Why didn’t you include a control group?

Also, it’s unclear whether the first 3-week training block or the second one was more effective—don't you think this should have been assessed? Wouldn’t it have been appropriate to include a mid-point test, at least for jump or sprint performance, to evaluate possible changes during the intervention?

You used 0.30 m/s to estimate 1RM, based on Sánchez-Medina’s study, which included only male participants. However, there are specific articles that focus on 1RM prediction in female populations, such as Pareja-Blanco et al. (https://doi.org/10.1055/a-1171-2287) and Nieto-Acevedo et al. (https://doi.org/10.3390/ijerph20064888). Wouldn't it be more appropriate to use a velocity threshold validated for women?

The prediction section is quite dense and difficult to follow. What practical application do you see for this analysis?

Round 2

Reviewer 2 Report

Comments and Suggestions for Authors

The authors adequately addressed my comments.

Reviewer 3 Report

Comments and Suggestions for Authors

The suggested revisions to the text have been thoroughly implemented, ensuring clarity, coherence, and accuracy of content. The material meets all required quality standards and is ready for publication without further revisions.